# Genomic Surveillance of *Salmonella* from the Comunitat Valenciana (Spain)

**DOI:** 10.3390/antibiotics12050883

**Published:** 2023-05-09

**Authors:** Andrea Sánchez-Serrano, Lorena Mejía, Maria Luisa Camaró, Susana Ortolá-Malvar, Martín Llácer-Luna, Neris García-González, Fernando González-Candelas

**Affiliations:** 1Joint Research Unit “Infection and Public Health”, FISABIO-University of Valencia, 46020 Valencia, Spain; andrea.sanchez@fisabio.es (A.S.-S.); mmejia@usfq.edu.ec (L.M.); 2Institute for Integrative Systems Biology (I2SysBio), CSIC-University of Valencia, 46980 Valencia, Spain; 3Instituto de Microbiología, Colegio de Ciencias Biológicas y Ambientales, Universidad San Francisco de Quito, Quito 170901, Ecuador; 4Public Health Laboratory, 46020 Valencia, Spainortola_sus@gva.es (S.O.-M.);; 5CIBER in Epidemiology and Public Health, 28029 Madrid, Spain

**Keywords:** *Salmonella enterica*, genomic surveillance, whole-genome sequencing, serotyping, molecular epidemiology, antimicrobial resistance

## Abstract

*Salmonella enterica* subspecies *enterica* is one of the most important foodborne pathogens and the causative agent of salmonellosis, which affects both humans and animals producing numerous infections every year. The study and understanding of its epidemiology are key to monitoring and controlling these bacteria. With the development of whole-genome sequencing (WGS) technologies, surveillance based on traditional serotyping and phenotypic tests of resistance is being replaced by genomic surveillance. To introduce WGS as a routine methodology for the surveillance of food-borne *Salmonella* in the region, we applied this technology to analyze a set of 141 *S. enterica* isolates obtained from various food sources between 2010 and 2017 in the Comunitat Valenciana (Spain). For this, we performed an evaluation of the most relevant *Salmonella* typing methods, serotyping and sequence typing, using both traditional and in silico approaches. We extended the use of WGS to detect antimicrobial resistance determinants and predicted minimum inhibitory concentrations (MICs). Finally, to understand possible contaminant sources in this region and their relationship to antimicrobial resistance (AMR), we performed cluster detection combining single-nucleotide polymorphism (SNP) pairwise distances and phylogenetic and epidemiological data. The results of in silico serotyping with WGS data were highly congruent with those of serological analyses (98.5% concordance). Multi-locus sequence typing (MLST) profiles obtained with WGS information were also highly congruent with the sequence type (ST) assignment based on Sanger sequencing (91.9% coincidence). In silico identification of antimicrobial resistance determinants and minimum inhibitory concentrations revealed a high number of resistance genes and possible resistant isolates. A combined phylogenetic and epidemiological analysis with complete genome sequences revealed relationships among isolates indicative of possible common sources for isolates with separate sampling in time and space that had not been detected from epidemiological information. As a result, we demonstrate the usefulness of WGS and in silico methods to obtain an improved characterization of *S. enterica enterica* isolates, allowing better surveillance of the pathogen in food products and in potential environmental and clinical samples of related interest.

## 1. Introduction

The genus *Salmonella* is classified into different serovars or serotypes on the basis of surface antigens according to the White–Kauffmann–Le Minor scheme [1]. This scheme has been a useful and efficient way to subtype *Salmonella*, facilitating international communication and comparison. Nevertheless, this typing scheme does not always correctly reflect the evolutionary relationships among the different strains of *S. enterica* subsp. *enterica*. On the one hand, most serotypes are polyphyletic, deriving from multiple independent ancestors [2]. On the other hand, homologous recombination and chromosomal mutations have resulted in genotypically and phenotypically diverse subtypes evolved from the same serovars [3]. Furthermore, serotyping has been traditionally performed by serological tests. This is a laborious technique, and the recognition of the wide range of serovars that abound in food products requires a wide panel of antigens that is not available in many laboratories. Moreover, it may lead to imprecise results [4].

Serotyping by serology has been progressively complemented or replaced by molecular typing methods, such as multi-locus sequence-based typing (MLST) [5,6,7,8]. However, these methods do not have enough discriminatory power to differentiate among closely related strains [8,9]. This limitation renders them insufficient for many epidemiological studies.

High-throughput sequencing (HTS) technologies have allowed rapid and affordable analyses of complete genomes [10]. Whole-genome sequences (WGS) allow an unprecedented level of discrimination among genetically related isolates, which can be applied to many interesting questions, such as accurate phylogenetic and phylogenomic analyses [11]. Moreover, WGS also facilitates the examination of serotype- or subtype-determining genes [12]. This in silico typing approach is being accepted and used for source attribution and epidemiological surveillance [13]. Several analytical approaches have been developed towards this goal [14,15,16], and various studies have shown the potential of WGS in epidemiological investigations [17,18].

The increasing use, often abuse, of antibiotics, especially for human health and for farm animals, has contributed to an increase in the prevalence of antimicrobial resistance. AMR isolates are found in many different environments, from the clinic to wastewaters and farms, which has prompted actions at many different levels and targets [19]. A reduction in the overuse of antibiotics in human health and farming and aquaculture is one of these, and a much closer monitoring of the genetic determinants of resistance is another example. Antimicrobial resistance is recognized as one of the major threats for human health for international organizations, such as WHO, FAO, and OECD [20]. This has led to the implementation of surveillance of antimicrobial resistance in human pathogens using WGS, especially for bacteria in the ESKAPE group (*Enterococcus faecium*, *Staphylococcus aureus*, *Klebsiella pneumoniae*, *Acinetobacter baumannii*, *Pseudomonas aeruginosa*, and *Enterobacter* spp.) and other major pathogens, such as *Mycobacterium tuberculosis* or highly pathogenic strains of *Escherichia coli*. The ECDC [21] and the WHO [22] have recommended the use of WGS as the gold standard methodology for surveillance of bacterial pathogens.

Access to the WGS of bacterial isolates also allows performing additional analyses that are not considered in routine surveillance. Among these, evaluating the presence of genetic determinants of antimicrobial resistance is one of the most relevant, especially under the One Health paradigm [23]. This is becoming the *de facto* main framework for the surveillance and control of antimicrobial resistance. Food-related pathogens are key players under this framework [24,25], and using genome information to identify new resistance determinants and monitor the spread of those already known is a major improvement over previous methodologies [26,27]. However, its implementation must consider not only the technological issues but also the subsequent analyses and the comparison with the regular methods used in each laboratory [28].

Here, we derived whole genome sequences of 141 *Salmonella* spp. isolates derived from public health surveillance studies of *Salmonella* in food products at the Comunitat Valenciana (Spain) from 2010 until 2017. We have obtained the sequence type and serotype of these isolates by traditional methods (Sanger sequencing and serology) and in silico. For in silico serotyping, we used two different programs, SISTR and SeqSero2, which are considered the most reliable open-source tools for in silico serovar prediction of *Salmonella*. In addition, we used the genomic information to determine the presence of antimicrobial resistance genes and to evaluate the minimum inhibitory concentrations (MICs) of a panel of antibiotics. Finally, we performed a detailed study of the relationships among them in order to detect possible clusters and transmissions.

## 2. Results

### 2.1. Quality of Sequencing Reads, Genome Assembly, and Annotation

We obtained complete genome sequences from 141 isolates (Appendix A). After filtering, the average number of reads per isolate decreased from 777,649.2 to 675,013.9, and their average size increased from 139.7 to 143.8 bp (Appendix A). The assemblies resulted in an average of 116.6 contigs per isolate and a genome size of 4,868,712 bp, with an average N50 value of 216,087.2 (Appendix A).

From the 141 initial isolates, we removed 6 from further analysis, because they were not identified as *Salmonella enterica* by Kraken (*n* = 3) or because they did not attain the minimum values of assembly quality (*n* = 3). The 135 genomes retained for further analyses had an average of 71.3 contigs (range 29–249).

### 2.2. Comparison between Phenotypic and In Silico Typing

The 135 isolates retained were tested by both serology and in silico serotyping with two different widely used programs, SISTR and SeqSero2. Serology resulted in 23 different serotypes, with serotype Typhimurium being the most abundant (36.6%). We could not determine the serotype in four isolates (Appendix A).

Both in silico serotyping programs resulted in 25 different serotypes and assigned the same serotype in 132 of the 135 isolates (97.8%) (Appendix A). The remaining three isolates (Se_V_139, Se_V_141 and Se_V_142) could not be classified by SISTR because it could not identify the O antigen (Table 1).

For these discordant cases, serology results did not agree with those obtained in silico (Se_V_139) or did not provide results (Se_V_141 and Se_V_142). Moreover, for Se_V_141, SeqSero2 and SISTR identified different antigenic factors for the *fljB* gene, which encodes the H2 antigen. In this sample, this gene showed many differences with the corresponding antigenic factor in both databases, which led us to suspect that a recombination event involving the *fljB* gene might have occurred.

The serology results were not coincident with the in silico serotyping tools for only two isolates (1.48%): Se_V_139, discussed above, and Se_V_3. Both in silico tools identified the latter as Paratyphi B, while it was assigned to serotype Schleissheim by serology. Both serotypes share the same antigenic factor for the *fliC* gene, but the Schleissheim serotype is monophasic, while Paratyphi B is biphasic and, therefore, has an additional *fljB* gene. The *fljB* gene was detected by both in silico tools, with a maximum score of 100.

The most common ST in our sampling was ST34 (38 isolates, 28.1%), followed by ST198 (14, 10.4%). Of the 30 STs obtained, 15 were represented by only 1 isolate. All the STs detected, except ST5689, were associated with an eBURST Group (eBG) (Appendix A). The STs obtained by Sanger sequencing and MLST analysis of WGS sequences agreed in 124 isolates (91.9%). Six of the eleven isolates in which discrepancies were found belonged to the Typhimurium serotype, and the different STs obtained by each method differed in a single allele (ST19 and ST34, 1 SNP). For the remaining five isolates, an ST could be assigned only by WGS.

MLST and serology definitions of the population lineages were highly similar. Each ST was normally associated with a serotype, except for three serotypes (Cerro, Derby, and Typhimurium) that presented more than one ST (Table 2) and a polyphyletic distribution.

### 2.3. Core Genome and Phylogenetic Analysis

In order to study possible transmission clusters and/or common sources among the isolates, we performed a phylogenetic study. In addition to the 135 isolates sequenced in this work, we included 308 additional *S. enterica* assemblies from RefSeq belonging to 19 of the serotypes identified in our dataset. However, some of the less common serotypes (*n* = 9) were left without representatives.

The pangenome of the dataset resulted in 11,024 different genes, and we obtained a core genome of 2225 genes. The resulting multiple alignment contained 2,078,455 nt and 136,511 variable sites. A maximum likelihood tree was reconstructed using the evolutionary model GTR + F + R3. The phylogenetic tree revealed that the different STs and serotypes found in this study are well-defined by long branches and high bootstrap support values (>90%) (Figure 1).

### 2.4. Antimicrobial Resistance Analysis

Antimicrobial resistance was studied by two approaches, the identification of known genetic determinants using STARAMR and predicting the MIC with machine learning. The two approaches yielded very similar results.

According to the predictions obtained by STARAMR (Table 3 and Appendix A), 29 of the 135 isolates (21.48%) did not carry any resistance determinant (genes or point mutations) to any antibiotic family. For the remainder, we found resistance to at least one antibiotic. In total, 39 different resistance genes were detected. Resistance to tetracycline was the most prevalent (73 isolates), followed by sulfisoxazole (70 isolates) and streptomycin (70 isolates). In addition, the analysis revealed 4 different point mutations in the gene *gyrA* (S83F, S83Y, D87N, D87Y) in 38 isolates. These mutations confer resistance to ciprofloxacin and nalidixic acid. The predicted MIC values revealed a high proportion of resistance against streptomycin (112 isolates), tetracycline (72 isolates), and ampicillin (46 isolates) (Table 3 and Appendix A). Only 18 isolates (13.3%) showed susceptible predicted MICs to any antibiotic.

Resistance determinants were not distributed evenly among serotypes (Figure 2). Serotype Chester is characterized by six resistance genes common to all its isolates, which are not present in other close serotypes. For the Rissen serotype, inferences of antibiotic sensitivity from the presence of ARGs or MIC values reflect important variation among the samples. Isolates from the Infantis serotype show two resistance patterns: while five isolates do not have any resistance gene, and seven isolates share the *ant(3″)- Ia*, *sul1* and *tet(A)* genes. This also occurs in the Derby serotype, with the resistance genes *aadA2*, *dfrA12*, *sul3*, and *tet(A)*. Moreover, the monophasic variant of *S. typhimurium* is related with 4 resistance genes and several isolates to predicted phenotypic resistance to sulfisoxazole (8 isolates) and ampicillin (15 isolates). For Senftenberg, Agona, Enteritidis, Newport, and most isolates of the Typhimurium serotype, no resistance genes were detected. Remarkably, the only *S. mikawasima* isolate of this study was predicted to be resistant to ceftriaxone, a first-choice antibiotic used to treat human salmonellosis.

Interestingly, we detected the *mcr-1* gene, conferring resistance to colistin in the only ST334 and Brandenburg serotype isolate analyzed. The gene was located in an IncX4 conjugative plasmid with no additional resistance genes. We found that the full plasmid was assembled in one single contig. This plasmid had a 99% identity and 100% coverage with the *S. enterica* strain CFSAN064033 plasmid pGMI17-001_2 (accession NZ_CP028174.1). A putative relaxase ORF1_3 was found at position 21,365–22,594, with an identity of 38.39%. Appendix A shows the sequence and structural comparison between pGMI17-001_2 and the contig.

### 2.5. Cluster Definitions and Epidemiological Investigations

Using the core genome alignment, we clustered the isolates in high-similarity clusters (HSC) of 10, 5, or 3 pairwise SNPs distances. Of the 135 samples, 82 were included in one of the HSC identified. We identified 26, 26, and 24 clusters with two or more isolates for thresholds of 10 (Clusters C1–C26), 5 (70 isolates, C27–C81), and 3 SNPs (62 isolates, C65–C88), respectively (Appendix A). All the samples grouped in the same cluster belong to the same ST and serotype (Figure 3).

As expected, most isolates that grouped in HSC also shared consistent epidemiological information (Appendix A). However, we found some cases of isolates included in the same HSC cluster with reported different epidemiological data (Table 4). Most discrepancies correspond to different types of samples in the same or close locality and year. However, we also detected one case in which four highly similar isolates were sampled through 3 years in different localities, although all the samples were related to poultry products.

In most isolates in HSCs, we detected AMR determinants (66/82, 80.5%) which resulted in predicted resistance to one or more antibiotics. Similarly, the inferred MIC values for these isolates led to predicted resistance to at least one antibiotic for the same total number of isolates, but there were several cases of discrepancy between the two methods. In fact, sensitivity to all tested antibiotics was inferred in only eight isolates.

A more detailed analysis of the inferred sensitivity profiles within HSC revealed additional discrepancies. Most of these were detected at the highest clustering level (10 SNPs), with 8 cases in which at least 1 of the isolates differed in the inferred resistance profile from ARGs (*n* = 5) and/or CMIs (*n* = 6). Some of these discrepancies were also observed at the 5 SNPs (*n* = 6) and even the 3 SNPs (*n* = 6) clustering levels (Appendix A).

## 3. Discussion

Applying the One Health approach to the surveillance and control of antimicrobial resistance requires the implementation of methodologies that can provide comparable results in different settings and for samples of different sources [29]. Traditionally, surveillance has been organism-driven, focusing on genotypic and/or phenotypic features that are more relevant for each pathogen. This led to the establishment of many different methodologies with different problems of reproducibility, transferability, standardization, and scalability across laboratories and pathogens. Some of these problems were minimized through strict protocols and regular quality and harmonization controls, apart from the use of common controls and platforms for sharing results. PulseNet was the best example of how much this strategy could achieve [30].

However, the introduction of high-throughput sequencing and the possibility of obtaining complete or near-complete genome sequences of bacterial isolates at reasonable costs and time has led to a dramatic change in how to achieve surveillance goals and solve most of the aforementioned problems. This has not been fulfilled completely yet, but the path seems clear, and most agencies are shifting to genome-based surveillance. Among the many advantages that this methodology offers, those that allow connecting the results or surveillance from different levels, sites, and species are especially relevant for the One Health approach.

In this work, we obtained a high level of agreement (98.52%) between in silico determined serotypes and those determined by serology. This level of agreement is higher than that obtained previously with a larger sample size [11]. In one of the cases of disagreement, the discordance between the Paratyphi B and Schleissheim serotypes is likely due to a lack of detection of agglutination for the H2 antigen, as suggested by the phylogenetic tree that places this isolate in a clade formed by the serotype Paratyphi B. In the other case, a completely different antigenic factor of the *fliC* gene was identified phenotypically and in silico.

For in silico serotyping, we employed two of the most reliable tools currently available, SeqSero2 and SISTR [31]. They reported a coincident result in 97.8% of the isolates (132/135). However, in two cases, SISTR could not determine one of the antigenic factors, while SeqSero2 unambiguously identified them. The serotype reported by SeqSero2 usually corresponds to the ST assigned to these isolates. However, due to the low number of representative genomes of the serotype assigned in the RefSeq database, SeqSero2 prediction could not be confirmed by core genome analysis.

In most isolates, the results of the MLST analysis performed by the two methods were consistent. However, the allele combination obtained by Sanger-based MLST did not correspond to known STs in five of the isolates, while by complete genome-based MLST, we could assign an ST to all of them. It is likely that the STs currently determined in four of them were not established in the database when these isolates were typed (between 2014 and 2017). The other discordant isolate presented STs with only one different allele. Their location in the phylogenetic tree suggests that the ST obtained from complete genomes is correct.

Although the results of the resistance genes analysis show that the isolates have a high content of resistance genes to several antibiotic families, this does not necessarily correspond to clinical resistance, as shown by the results of the MIC predictor (Figure 1). In addition, for several of the predicted resistant isolates, especially those resistant to streptomycin, no resistance determinants were identified. Similarly, we found an isolate resistant to ceftriaxone, but we have not detected any resistance gene or mutation that could originate it. This fact can be explained by the presence of not-detected genes related to resistance to these antibiotics. It is necessary to have more samples to investigate the genetic determinant that confers resistance to this antibiotic.

For the remaining isolates, the predicted phenotypes do not involve antibiotics currently used in the treatment of salmonellosis, so our isolates are not clinically relevant. However, we found the *mcr-1* gene in the only study isolate classified as Brandenburg serotype and ST334. This gene is responsible for the colistin resistance. It has been identified worldwide in *Salmonella* isolates obtained from humans, animals, the environment, and food, but as far as we know, it has not been reported in the Brandenburg serotype. The IncX4 plasmid that carries the gene *mcr-1* has been detected in *Salmonella* food isolates belonging to other serotypes in different countries [32,33,34]. In Spain, this plasmid has been found in *Escherichia coli* isolates [35,36], but to our knowledge, it has not been reported in *Salmonella*.

Colistin has been widely used in animal production in several countries for prophylaxis and growth promotion [37]. In 2015, the use of colistin in swine production in Spain was estimated to be 50 mg/PCU (population correction units), but in the past decade, the spread of carbapenemase-producing *Enterobacteriaceae* led to re-implementing the use of colistin as a last therapeutic option for the treatment of human infections. For this reason, a reduction target of 5 mg/PCU was set for the next years [38]. Now, colistin is classified by The World Health Organization as an antibiotic of critical importance in human clinical settings [39]. Genetic analysis and epidemiological surveillance are crucial for minimizing the spread of colistin resistance, and early identification of colistin-resistant isolates in any setting is one of the most useful outcomes of WGS-based surveillance.

The phylogenetic reconstruction highlights the genetic heterogeneity of several serotypes [40,41,42]. In relation to our study isolates, each clade of the polyphyletic serotypes corresponds to a specific ST, except in the case of *S. typhimurium* and its monophasic variant. Although sequence typing generally allows very efficient lineage discrimination, it can group isolates as different from these (Figure 2). Furthermore, STs do not always correspond to ancestral relationships, as in the case of ST19, which has a paraphyletic distribution. On the other hand, MLST schemes lack the necessary resolution for adequate discrimination of outbreaks. However, epidemiological information in combination with genomic information and phylogenetic analysis can provide this higher level of discriminatory power, allowing the identification of closely related samples.

Using this information, we have been able to detect closely related isolates that may be associated with the same transmission source. High-similarity clusters were generally consistent with epidemiological data. However, for some of these clusters, the isolates were obtained from apparently unrelated products (Clusters C7, C18, and C65) and/or at distant dates (Cluster C6) (Appendix A). For all these cases, it would be convenient to study in more detail the production chain of the products and the context in which the samples were obtained.

The most obvious limitation of this work is the lack of information on sensitivity profiles to antibiotics, as these tests were not included in the routine procedures at the Public Health Laboratory. Lack of this information undermines our in silico evaluation of antimicrobial sensitivity profiles, but it also highlights one of the potential benefits of using WGS for surveillance of food-related products because this allows the early identification of pathogens with potential resistance to various antibiotics. In addition, the use of machine learning approaches to infer AMR profiles and MICs, such as PATRIC [43], is becoming popular, as these methods are revealing high accuracy and reliability for different pathogenic bacteria [44,45,46]. In any case, these predictions should be tested more extensively to rule out and/or establish limits to machine learning inferences before they can be used to adopt appropriate actions and implement recommendations for human and animal public health.

Another limitation is that we used a core genome approach to define HSCs, while several of these reveal differences in antimicrobial susceptibility and epidemiological features. It is possible that some of these HSCs will be further subdivided into separate groups should their accessory genomes be considered in the analysis, thus generating more coherent and congruent subgroups. Nevertheless, this problem is much more easily tackled with data from WGS analysis than with results from serological or direct multi-locus sequence typing, where resolving these discrepancies is not possible.

## 4. Materials and Methods

### 4.1. Sample Selection

We used a set of 258 *Salmonella* spp. isolates collected by the Public Health Laboratory of the Comunitat Valenciana (CV, Spain) between 2010 and 2017. This collection contains information for each isolate on serotype, sequence type (ST), and epidemiological data, such as sampling source, date, and location.

Serotyping was performed by slide agglutination with O and H antigen-specific sera according to the White–Kauffmann–Le Minor scheme [1], while STs were obtained following the Warwick University protocols (available at https://enterobase.readthedocs.io/en/latest/mlst/mlst-legacy-info.html).

We used epidemiological and serological data to select samples from the collection that were likely to be associated with possible transmission chains or serotype misidentification or belonged to infrequent serotypes. We selected 141 isolates following these criteria (Appendix A).

### 4.2. Whole-Genome Sequencing and Gene Annotation

DNA was extracted using EasyMag (Biomérieux, Lyon, France). Isolates were sequenced using Illumina NextSeq technology. Sequencing libraries were prepared with the Nextera XT DNA library preparation kit according to the manufacturer’s protocol. As a result, we obtained paired-end reads of 2 × 150 bp. Raw sequence data generated in this study were deposited in the European Nucleotide Archive (ENA) under project PRJEB49974.

FastQC and MultiQC were used to assess the quality of the raw reads. Short reads (<100 bp) or bad quality reads (mean quality < 25) were removed from ensuing analyses using PrinSEQ [47]. To discard contaminated or misidentified isolates, we examined the taxonomic content of each sample using Kraken [48]. Cleaned and filtered sequence reads were assembled de novo using SPAdes [49]. Assembly quality was evaluated using QUAST [50]. Assemblies with more than 600 contigs or a GC content lower than 50% or higher than 52% were discarded from subsequent analyses. Gene annotation was performed using PROKKA [51].

### 4.3. In Silico Isolate Characterization: Serotypes, STs, and AMR

We used seq2MLST v. 1.0.1 [52] to identify the STs of each sample. For in silico serotyping, we compared the results of SeqSero2 [15] and SISTR [16].

Antibiotic resistance genes, mutations, and plasmids were identified using STARAMR 0.7.2 (https://github.com/phac-nml/staramr), which uses the ResFinder [53], PointFinder [54], and PlasmidFinder [55] databases. This tool provides a drug resistance prediction based on gene identification, but it does not imply clinical resistance. For this reason, we used a method based on machine learning developed by the PathoSystems Resource Integration Center (PATRIC) [43] to predict the minimum inhibitory concentration (MIC) for 15 clinically important antibiotics. Table 5 shows the breakpoints used to consider resistance.

### 4.4. Core Genome and Evolutionary Analysis

All publicly available complete *S. enterica* subsp. *enterica* genome assemblies were retrieved from RefSeq on 29 June 2020. We used SISTR and SeqSero2 to determine the serotype of each genome. We only kept those genomes of serotypes represented in our sample collection. In addition, we removed genomes with a discrepancy between the serotype indicated at the NCBI and the one obtained using SISTR and SeqSero2.

To analyze the relationships among the different isolates, we studied their pangenome. Shared orthologous genes were determined with Proteinortho5 [56]. Public genomes that did not contain most of the genes present in the study samples were discarded. In addition, duplicate genes were removed from the pangenome. For each isolate, genes in the strict core genome were concatenated. We used AMAS [57] to remove genes with an abnormally large number of undetermined characters or a high proportion of variable sites.

To reconstruct the global phylogeny of the isolates, we used IQTREE [58] employing the ultrafast bootstrap option [59] with 1000 replicates. Finally, MEGAX [60] was used to obtain the distance matrix among the analyzed genomes. To represent the different trees and include the serotyping results, we used iTOL (https://itol.embl.de).

### 4.5. Cluster Definition and Epidemiological Investigation

With the aim of identifying clusters that were more likely to be associated with the same contamination source, we used the core genome alignment to group the isolates into clusters of increasing levels of similarity [61]. For this, we performed single-linkage clustering of the SNP distance matrix at three threshold levels (10, 5, and 3 SNPs), i.e., the maximum length separating the sequences in the same cluster. Clustering was performed using the R library DECIPHER [62]. Moreover, we related these clusters with epidemiological information (source/product, geographic location, and collection date) and predicted antimicrobial sensitivity (see above) for each isolate.

## 5. Conclusions

This study shows the potential of complete genome sequencing in epidemiological surveillance of *S. enterica* and highlights the need to incorporate WGS and bioinformatic techniques into routine surveillance analyses. As we have shown, an in silico approach is advantageous for the identification of these serotypes. However, the addition of complete genomes sequences of the less prevalent serotypes in *Salmonella* databases is essential for improving the quality of in silico methods for typing *Salmonella* isolates and resolve ambiguities and misassignments derived from serology.

## Figures and Tables

**Figure 1 antibiotics-12-00883-f001:**
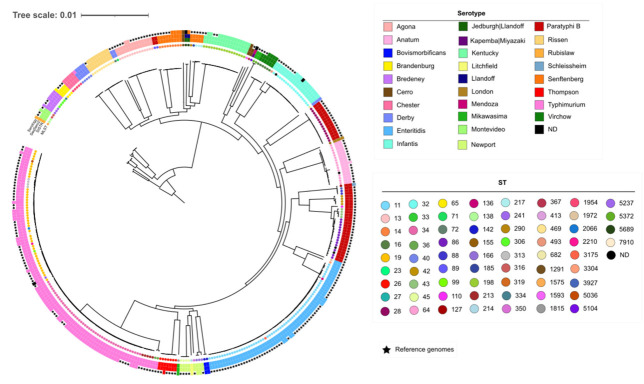
Maximum likelihood phylogenetic tree from the strict core alignment of the 135 studied isolates and 235 reference genomes. All the internal branches have bootstrap values larger than 90%. Inner dots indicate the ST. The circles around the tree show the serotypes inferred by SISTR (inner circle), SeqSero2 (middle circle), and the serology (outer circle). (Accessible at https://itol.embl.de/tree/1957716215329871633951913).

**Figure 2 antibiotics-12-00883-f002:**
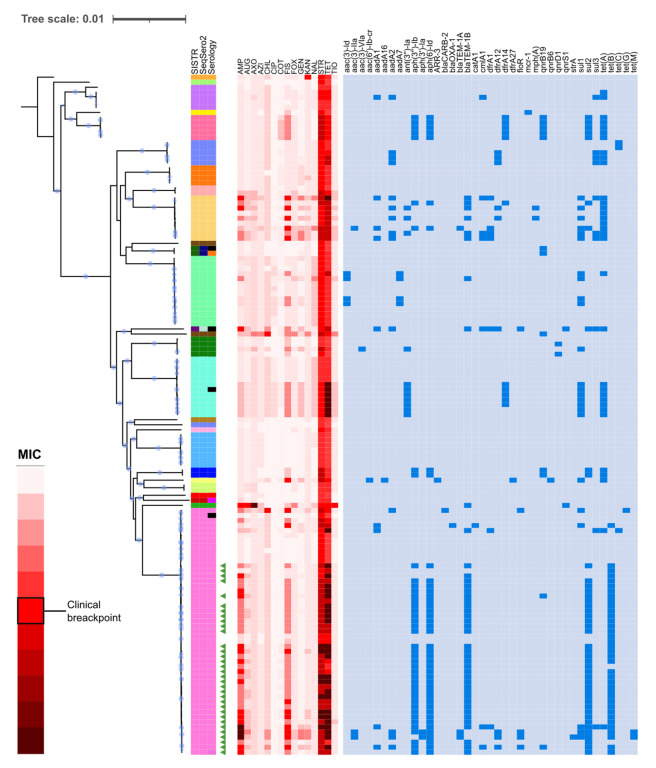
Relationships between the phenotypic and genotypic resistance determinants in the different serotypes. The red heat-map indicates the MIC predicted for each isolate. In the blue heat-map, the resistance genes found in each serotype are indicated. AMP: Ampicillin, AUG: Augmentin, AXO: Ceftriaxone, AZI: Azithromycin, CHL: Chloramphenicol, CIP: Ciprofloxacin, COT: Co-trimoxazole, FIS: Sulfisoxazole, FOX: Cefoxitin, GEN: Gentamicin, KAN: Kanamycin, NAL: Nalidixic acid, STR: Streptomycin, TET: Tetracycline, and TIO: Ceftiofur. Accessible at: https://itol.embl.de/tree/1957716215415541614592390.

**Figure 3 antibiotics-12-00883-f003:**
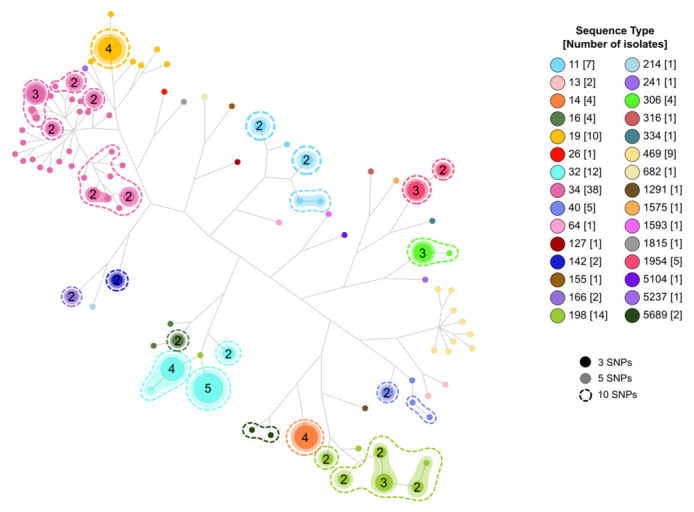
Network analysis showing the relationships among isolates and STs and the clusters obtained according to the different threshold levels. Numbers in circles represent identical isolates.

**Table 1 antibiotics-12-00883-t001:** Serology results and in silico determination of the antigens by SeqSero2 and SISTR for the isolates with discordant results between serology and in silico or between in silico serotyping programs. ND: Not Determined.

		Se_V_3	Se_V_139	Se_V_141	Se_V_142
**Serology**		Schleissheim	Senftemberg	ND	ND
**SISTR cgMLST**		-	Kentucky	Hadar	Kentucky
**SISTR antigens**	**Serotype**	Paratyphi B	Jedburgh|Llandoff	Kapemba|Miyazaki	Jedburgh|Llandoff
	**O antigen**	1, 4, 5, 12	ND	ND	ND
	**H1 antigen**	b	z29	l, v	z29
	**H2 antigen**	1, 2	-	1, 7	-
**SeqSero2**	**Serotype**	Paratyphi B	Llandoff	Mendoza	Llandoff
	**O antigen**	4	1, 3, 19	9	1, 3, 19
	**H1 antigen**	b	z29	l, v	z29
	**H2 antigen**	1, 2	-	1, 2	-

**Table 2 antibiotics-12-00883-t002:** Number of isolates and STs of each serotype identified in the dataset (135 isolates from CV and 235 from the RefSeq database). Undetermined STs (*n* = 3) are not included.

Serotype	Number of Isolates		Number of STs		STs	
	Study Isolates	RefSeq Genomes	Study Isolates	RefSeq Genomes	Study Isolates	RefSeq Genomes
Agona	2	12	1	1	ST13	ST13
Anatum	1	15	1	1	ST64	ST64
Bovismorbificans	2	0	1	-	ST142	-
Brandenburg	1	3	1	1	ST334	ST65
Bredeney	5	2	2	1	ST241	ST241
Cerro	2	1	2	1	ST1291, ST1593	ST367
Chester	5	0	1	-	ST1954	-
Derby	6	2	2	2	ST40, ST682	ST71, ST72
Enteritidis	7	57	1	5	ST11	ST11, ST3175, ST136, ST1972, ST3304
Infantis	12	8	1	2	ST32	ST32, ST493
Kentucky	14	3	1	1	ST198	ST198
Litchfield	1	0	1	-	ST214	-
Llandoff	2	0	1	-	ST5689	-
London	1	0	1	-	ST155	-
Mendoza	1	0	1	-	ST5104	-
Mikawashima	1	2	1	1	ST1815	ST5372
Montevideo	1	3	1	2	ST316	ST138, ST316
Newport	2	6	1	2	ST166	ST45, ST350
Paratyphi B	1	45	1	8	ST127	ST28, ST110, ST42, ST86, ST89, ST43, ST88, ST3927
Rissen	9	1	1	1	ST469	ST469
Rubislaw	1	0	1	-	ST1575	-
Senftenberg	4	10	1	4	ST14	ST14, ST217, ST290, ST185
Thompson	1	6	1	1	ST26	ST26
Typhimurium	49	57	3	10	ST34, ST5237, ST19	ST19, ST36, ST34, ST313, ST2210, ST213, ST2066, ST5036, ST99, ST7910
Virchow	4	2	1	1	ST16	ST16

**Table 3 antibiotics-12-00883-t003:** Number of isolates predicted as resistant by in silico MIC determination and by STARAMR. Genes involved in resistance detected by STARAMR are also shown.

	MIC Predictor	STARAMR	
Antibiotic	Number of Resistant Isolates	Number of Resistant Isolates	Genes Detected
**Streptomycin**	112 (82.96%)	70 (51.85%)	*aadA7*, *aadA16*, *aadA1*, *aadA2*, *ant(3″)-Ia*, *aph(3″)-Ib*, *strA*
**Tetracycline**	72 (53.33%)	73 (54.07%)	*tet(A)*, *tet(B)*, *tet(C)*, *tet(G)*, *tet(M)*
**Ampicillin**	20 (14.81%)	49 (36.30%)	*blaTEM-1B*, *blaTEM-1A*, *blaCARB-2*, *BlaOXA-1*
**Ciprofloxacin**	0	49 (36.29%)	*aac(6′)-Ib-cr*, *qnrB6*, *qnrB19*, *qnrD1*, *QnrS1*
**Sulfisoxazole**	16 (11.85%)	70 (51.85.74%)	*sul1*, *sul2*, *sul3*, *dfrA12*
**Chloramphenicol**	2 (1.48%)	11 (8.15%)	*cmlA1*, *floR*, *CatA1*
**Ceftriaxone**	1 (0.74%)	NA	
**Kanamicina**	1 (0.74%)	44 (32.59%)	*aph(6)-Id*, *aph(3′)-Ia*
**Ceftiofur**	1 (0.74%)	NA	
**Augmentin**	2 (1.48%)	NA	
**Azithromycin**	0	NA	
**Co-trimoxazole**	0	NA	
**Cefoxitin**	0	NA	
**Gentamicin**	0	9 (6.67%)	*aac(3)-Id*, *aac(6′)-Ib-cr*, *aac(3)-Via*, *aac(3)-Iia*
**Nalidixic acid, Ciprofloxacin**	0	49 (36.30%)	*gyrA*
**Colistin**	NA	1 (0.74%)	*mcr-1*
**Rifampicin**	NA	1 (0.74%)	*ARR-3*
**Trimethoprim**	NA	28 (36.30%)	*sul1*, *sul2*, *sul3*, *dfrA12*
**Erythromycin, Azithromycin**	NA	2 (1.48%)	*mph(A)*

**Table 4 antibiotics-12-00883-t004:** Detected clusters for which epidemiological data are not consistent with genetic similarity.

	Isolate	Serology	SeqSero2	SISTR	Cluster (3 SNPs)	Cluster (5 SNPs)	Cluster (10 SNPs)	ST	Product	Place	Year
**Different sample sources**	Se_V_62	Typhimurium	Typhimurium	Typhimurium	C7	C48	C84	19	Egg	Enguera	2016
	Se_V_64	Typhimurium	Typhimurium	Typhimurium	C7	C48	C84	19	Humboldt squid	Ayora	2017
	Se_V_65	Typhimurium	Typhimurium	Typhimurium	C7	C48	C84	19	Octopus	Torrent	2017
	Se_V_84	Typhimurium	Typhimurium	Typhimurium	C7	C48	C84	19	Ice cream	Alcalà de Xivert	2016
	Se_V_85	Enteritidis	Enteritidis	Enteritidis	C18	C59	C94	11	Salad	Valencia	2014
	Se_V_86	Enteritidis	Enteritidis	Enteritidis	C18	C59	C94	11	Chicken	Valencia	2014
	Se_V_87	Enteritidis	Enteritidis	Enteritidis	C26	C65	C99	11	Oysters	Valencia	2017
	Se_V_89	Enteritidis	Enteritidis	Enteritidis	C25	C65	C99	11	Pork	Gandia	2017
**Distant sampling times**	Se_V_53	Infantis	Infantis	Infantis	C6	C47	C83	32	Chicken	Valencia	2017
	Se_V_71	Infantis	Infantis	Infantis	C6	C47	C83	32	Chicken	Gandía	2015
	Se_V_72	Infantis	Infantis	Infantis	C6	C47	C83	32	Chicken	Rafelbunyol	2016
	Se_V_49	Infantis	Infantis	Infantis	C6	C47	C83	32	Chicken	Rafelbunyol	2016

**Table 5 antibiotics-12-00883-t005:** Breakpoints used for antibiotic resistance testing. Breakpoints established by CLSI were used when available. There are no CLSI breakpoints for streptomycin or ceftiofur, so NARMS-established breakpoints (https://www.cdc.gov/narms/antibiotics-tested.html) were used instead.

Antibiotic	MIC Breakpoints (μg/mL)
Ampicillin (AMP)	≥32
Augmentin (AUG)	≥16
Ceftriaxone (AXO)	≥4
Azithromycin (AZI)	≥32
Chloramphenicol (CHL)	≥32
Ciprofloxacin (CIP)	≥1
Co-trimoxazole (COT)	≥4
Sulfisoxazole (FIS)	≥512
Cefoxitin (FOX)	≥32
Gentamicin (GEN)	≥16
Kanamicina (KAN)	≥64
Nalidixic acid (NAL)	≥32
Streptomycin (STR)	≥32
Tetracycline (TET)	≥16
Ceftiofur (TIO)	≥8

## Data Availability

Short-reads from sequencing experiments reported in this article are deposited at the European Nucleotide Archive under project PRJEB49974. Additional details of the phylogenetic trees shown in Figure 1 and Figure 2 are accessible through the links provided in the corresponding legends.

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
