# Peer review of "Genomic Surveillance of Salmonella from the Comunitat Valenciana (Spain)"

_antibiotics, 2023, doi:10.3390/antibiotics12050883_

Round 1

Reviewer 1 Report

In this manuscript, Andrea Sánchez-Serrano and Lorena Mejía showed the potential of complete genome sequencing in epidemiological surveillance of S. enterica and highlights the need to incorporate WGS and bioinformatic techniques into routine surveillance analyses.

In this paper, they have applied whole-genome sequencing (WGS) technology to analyze a set of 141 S. enterica isolates obtained from different food sources between 2010 and 2017 in the Comunitat Va- 20 lenciana (Spain). They performed an evaluation of most relevant Salmonella typing methods, serotyping and sequence typing, using both traditional and in silico approaches. The result of this study demonstrated the usefulness of WGS and in silico methods to obtain an improved characterization of S. enterica enterica isolates, allowing better surveillance of the pathogen in food products and in potential environmental and clinical samples of related interest.

The strengths of this article are as followed: This work provides better detection techniques for Salmonella in food, as well as potential related environments and clinical samples.

Author Response

Thanks for your comments. We see no need to change the original manuscript after them.

Reviewer 2 Report

Reviewer Comments to the Authors

Although salmonellosis in detail, it can be most effectively prevented by a One Health approach that considers the entire chain from farm to fork. Such preventive and holistic approaches can reduce both the burden of disease on human health and the economic burden on developing economies and thus represent significant potential for improvement related to food safety, as seen in One Health.

The manuscript is of extraordinary importance for public health, highlighting new methods by which Salmonella infections can be controlled. The manuscript is well-written and structured, but some aspects still require corrections before being considered for publication, and I will detail these deficiencies in what follows.

Abstract

Lines 19-20: the name of the S. enterica species is written in italics without underlining

Line 25: what do the abbreviations AMR, SNP mean? They are used for the first time in the manuscript

Line 27: what does MLST entail? It is used for the first time in the text

Introduction

Lines 44-45: is the statement personal, or does it reflect the results of studies conducted? If it belongs to you, I recommend moving to the discussion chapter. Otherwise, it requires a bibliographic source.

Lines 58-59: require the insertion of the bibliographic source

Lines 69-71: I recommend rewording the sentence to make it more transparent and easier to follow. The overuse of antibiotics is more about the farm animals and not the farm.

Lines 80-81: I recommend inserting the bibliography.

Results

Lines 125, 126, 128, 132, 133, 134: Gene names are written in italics

Line 151: the name of the species S. enterica is written in italics

Line 168: what does STARAMR stand for? It is the first mention in the text.

Lines 173-176: would it be helpful to mention what strains resistance came from?

Line 175, 187, 203: Gene names are written in italics

Lines 177-179: would it be helpful to mention the strains of that resistance?

Line 189: Gene names are written in italics

Line 193: Gene names are written in italics

Line 207: Gene names are written in italics

Line 214: what does HSC stand for?

Discussions

Lines 248-249: bibliographic source

Lines 301-304: bibliographic source

Lines 327-329: Why is this type of analysis not included in the routine diagnosis of public health laboratories?

In the discussion chapter, this theme is widely debated. However, there is a lack of information about other similar research/studies on a global level that analyzed these techniques to have an overview of their usefulness and to bring solid arguments because of immediate implementation in human and veterinary clinical activity.

I recommend rewording the chapter.

Results

Line 385: the name of the species is written in italics

In this chapter, there are many abbreviations, which are sometimes challenging to follow, and most of them are described for the first time in the manuscript without providing the full name. I recommend a separate document that includes all the abbreviations and is easier to follow.

Conclusions

Lines 414-415: the name of the species is written in italics

I recommend reformulating and highlighting the importance of this study. I have possibly mentioned the disadvantages (financial or of another nature) for which these techniques are not included in the routine procedures of public health laboratories. Currently, this chapter is formulated very technically and does not offer the possibility to those not strictly specialized to understand the conclusions.

Author Response

Please see the atttachment

Reviewer 3 Report

This manuscript presents a very interesting and quite novel approach to one of the main issues in Food Safety: The use of WGS as a routine methodology for the surveillance of food-borne bacterial pathogens. By WGS, authors have performed genotyping of a set of S. enterica strains, comparing results with that of conventional typing methods. They have also detected antimicrobial resistance determinants and predicted minimum inhibitory concentrations. Finally, they have used WGS and epidemiological data to track possible contaminant sources and their relationship to AMR in the geographical area of the study.

The objectives of the work are ambitious but well delimited, methodology is adequate, and results are very interesting.

My only concern is about the main limitation of the work, also pointed out by authors: There is a lack of information on phenotypic sensitivity profiles to antibiotics, as tests were not performed. This fact has made impossible to compare the in silico with in vitro results.

To diminish the impact of this limitation on the work, authors should discuss (supporting their statements with the appropriate references about previous works) the effectiveness of STARAMR and, specially, the PATRIC method based on machine learning to predict the minimum inhibitory concentration (MIC) for antibiotics.

Equally, after paragraph 282-285 (“Although the results of the resistance genes analysis show that the isolates have a high content of resistance genes to several antibiotic families, this does not necessarily correspond with clinical resistance, as shown by the results of the MIC predictor (Figure 1). In addition, for several of the predicted resistant isolates, especially those resistant to streptomycin, no resistance determinants have been identified”), authors should add a comment about the possibility that the discordances among genetic determinants detected in strains and predicted resistance may be due to inaccuracies of the MIC predictor method used and that it should be necessary to perform a more exhaustive study, with in vitro MIC determinations, to definitively take a conclusion about this subject.

Round 2
